# Mendelian randomization reveals no correlations between herpesvirus infection and idiopathic pulmonary fibrosis

Haihao Yan[‡], Chenghua Zhu[‡], Xiao Jin, Ganzhu Feng[ORCID]*

Department of Respiratory Medicine, The Second Affiliated Hospital of Nanjing Medical University, Nanjing, China

‡ HY and CZ contributed equally to this work and share the first authorship.
* fgz62691@163.com

## Abstract

### Background

Previous studies have found that the persistence of herpesvirus significantly increases the risk of idiopathic pulmonary fibrosis (IPF), but it is unclear whether this effect is causal. We conducted a two-sample Mendelian randomization (MR) study to evaluate the causal relationship between three herpesvirus infections and IPF.

### Methods

We used genome-wide association studies (GWAS) data from three independent datasets, including FinnGen cohort, Milieu Intérieur cohort, and 23andMe cohort, to screen for instrumental variables (IVs) of herpesvirus infection or herpesvirus-related immunoglobulin G (IgG) levels. Outcome dataset came from the largest meta-analysis of IPF susceptibility currently available.

### Results

In the FinnGen cohort, genetically predicted Epstein-Barr virus (EBV) (OR = 1.105, 95%CI: 0.897–1.149, p = 0.815), cytomegalovirus (CMV) (OR = 1.073, 95%CI: 0.926–1.244, p = 0.302) and herpes simplex (HSV) infection (OR = 0.906, 95%CI: 0.753–1.097, p = 0.298) were not associated with the risk of IPF. In the Milieu Intérieur cohort, we found no correlations between herpesvirus-related IgG EBV nuclear antigen-1 (EBNA1) (OR = 0.968, 95% CI: 0.782–1.198, p = 0.764), EBV viral capsid antigen (VCA) (OR = 1.061, 95CI%: 0.811–1.387, p = 0.665), CMV (OR = 1.108, 95CI%: 0.944–1.314, p = 0.240), HSV-1 (OR = 1.154, 95%CI: 0.684–1.945, p = 0.592) and HSV-2 (OR = 0.915, 95%CI: 0.793–1.056, p = 0.225) and IPF risk. Moreover, in the 23andMe cohort, no evidence of associations between mononucleosis (OR = 1.042, 95%CI: 0.709–1.532, p = 0.832) and cold scores (OR = 0.906, 95% CI: 0.603–1.362, p = 0.635) and IPF were found. Sensitivity analysis confirmed the robustness of our results.

**Data Availability Statement:** All data are available on the aforementioned public repository and are accessible with permission from the corresponding data committee (https://www.ebi.ac.uk/gwas/downloads/summary-statistics). No restrictions on

data availability other than those imposed by the corresponding data committee.

**Funding:** The author(s) received no specific funding for this work.

**Competing interests:** The authors have declared that no competing interests exist.

## Conclusions

This study provides preliminary evidence that EBV, CMV, and HSV herpesviruses, and herpesviruses-related IgG levels, are not causally linked to IPF. Further MR analysis will be necessary when stronger instrument variables and GWAS with larger sample sizes become available.

## Introduction

Idiopathic pulmonary fibrosis (IPF) is a chronic pneumonia characterized by progressive and interstitial fibrosis. The healthy alveolar structure is destroyed, and the abnormal extracellular matrix replaces the normal tissue, resulting in decreased lung compliance, gas exchange disorders, and eventually respiratory failure and death [1]. IPF usually occurs in the elderly, its aetiology is unknown, and the median survival time after diagnosis is about 3 years [2]. In latest years, the incidence of IPF has been increasing yearly, and now the incidence is about 1.25–3.77 per 100000 per year [3]. IPF is incurable, and there is no effective treatment for IPF except lung transplantation [4].

For a long time, IPF has been suspected to be associated with viral and bacterial infections. The infection rate of herpesvirus in the European population is very high; in which the infection rate of herpes simplex virus (HSV) is as high as 95%, Epstein-Barr virus (EBV) is 90%, and cytomegalovirus (CMV) is about 60% [5]. The prevalence rate of the elderly is significantly higher than that of the young. Once the host is infected with the herpes virus, the infection lurks and lasts for a lifetime. The virus lurks in the nucleus, persists in the episomal form, and then periodically reactivates in the form of the lytic virus [5].

Previous studies found that EBV immunoglobulin G (IgG), CMV IgG, and HSV IgG titers in IPF patients were higher than in other lung diseases and normal controls, which suggests that some herpesviruses may be related to the development of IPF [6]. A recent large meta-analysis [7] also showed that the persistence of herpesviruses such as EBV, CMV, and human herpesvirus (HHV) 7 and 8 significantly increased the risk of IPF (p = 0.001), but not with the deterioration of IPF (p = 0.988). However, the current observational research has a variety of defects, such as residual and unmeasured confusion, detection deviation and reverse causality [8]. Moreover, IPF is a rare disease, so it is difficult to raise enough participants in longitudinal studies to fully explore the association between herpesvirus and IPF. Because of these limitations, better methods are needed to assess the causal effect of herpesvirus infections on pathogenesis of IPF.

Mendelian Randomization (MR) is a genetic epidemiological method for inferring causal relationships in disease risk factors [9–11]. MR utilizes genetic variation as instrumental variables (IVs) for gene-level causal inference. Because the specific alleles represented by genetic variation remain constant throughout an individual's lifetime [12], MR is, to a large extent, able to mitigate the influence of confusion factors and is less susceptible to reverse causation interference [13]. Although IVs may influence outcomes through confusion factors, leading to biased effect estimates, employing consistent estimation methods across various assumptions of pleiotropy can avoid this bias [14]. Recent large-scale infection, post-infection humoral immune response and IPF GWAS datasets enable us to use a two-sample MR design to assess the correlation between three herpesviruses (EBV, CMV and HSV) and the risk of IPF.

## Material and methods

### Study design

Fig 1 showed the flow diagram of the two-sample MR design. Our study adheres to the latest STROBE-MR presentation for reporting MR studies [15]. Specifically, the MR study needs to meet the following assumptions: First, IVs should be tightly correlated to exposure (herpesvirus infection and circulating IgG levels); Second, IVs are not correlated to any possible confounders; Third, IVs can only impact the outcome (IPF) through exposure, and there should be no other pathway. S1 Table summarized the data sets used in this study. The data in this study came from publicly available GWAS data. All the consortia that initially participated in the study have completed the participants' ethical approval and written informed consent.

### Exposure GWAS data set and single-nucleotide polymorphisms (SNPs) selection

For herpesvirus infections, we used GWAS summary statistics from the FinnGen study (https://www.finngen.fi/en/access_results). Herpesvirus infection is defined by the International Classification of Diseases (ICD) in Finland's outpatient, hospitalization and death registries. In the latest version, EBV infection (R8) included 2,099 cases and 333,715 controls; CMV infection (R7) included 428 cases and 301,439 controls; HSV infection (R8) included 2,924 cases and 330,087 controls (S1 Table).

IVs for circulating IgG levels are selected from summary statistics published on NHGRI-E-BIGWAS [16]. The sample is from the Milieu intérieur cohort, consisting of 1000 healthy people (500 females) recruited by BioTrial (Rennes, France). Serum-specific IgG was detected by BioPlex™ 2200 IgG kit for different herpesvirus antigens. The positive numbers of antibody levels related to this study among 1000 individuals were EBV nuclear antigen-1 (EBNA1) 914cases, EBV viral capsid antigen (VCA) 956cases, CMV 347cases, HSV-1 645cases, HSV-2

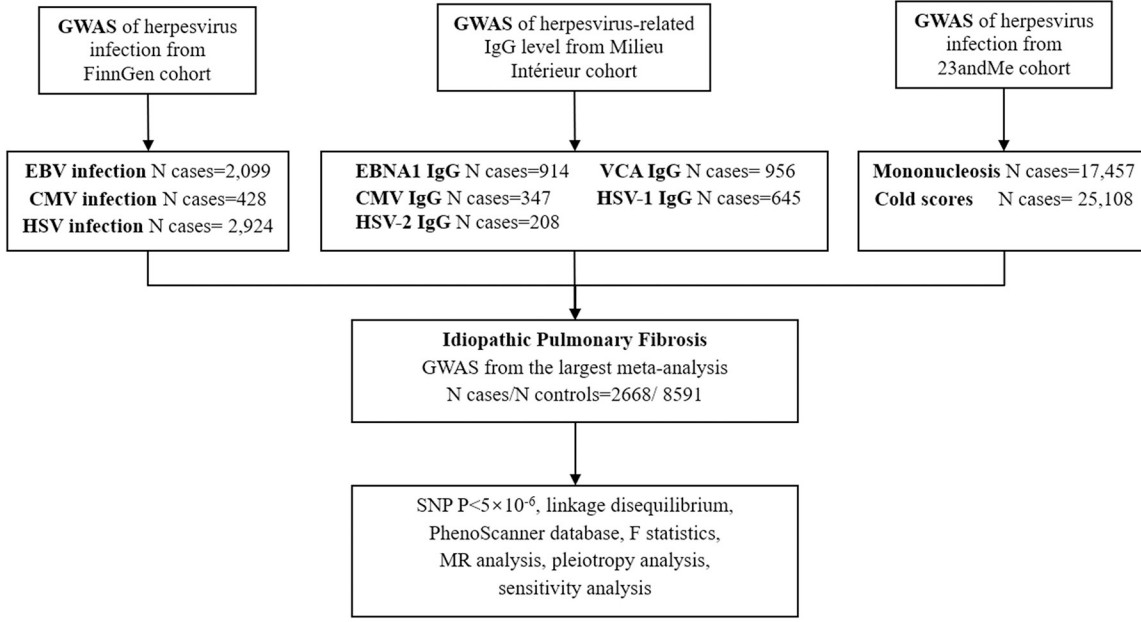

**Fig 1. The flow diagram of the process in this Mendelian randomization analysis.** GWAS, genome-wide association studies; SNP, single nucleotide polymorphism; EBV, Epstein-Barr virus; CMV, cytomegalovirus; HSV, herpes simplex; EBNA1, EBV nuclear antigen-1; VCA, EBV viral capsid antigen; IgG, immunoglobulin G.

208cases (S1 Table). After log10 transformation of IgG level, additive regression analysis with adjusted age, sex, total IgG and the first two principal components as covariates was used to analyze genetic association.

We further analyzed the GWAS summary statistics from the 23andMe cohort [17]. In this GWAS analysis, a strict self-reported questionnaire about their history of infection was used to define phenotypes [17]. Participants with >97% European ancestry were selected through local ancestry analysis. We selected mononucleosis (17457 cases and 68446 controls) and cold scores (25108 cases and 63332 controls) as exposure data, which were caused by EBV and HSV, respectively (S1 Table).

Consistent with previous studies [18, 19], SNPs were selected for analysis at a relatively liberal threshold ($p<5\times10^{-6}$) to obtain a sufficient number of SNPs. Then, we pruned the SNPs with horizontal pleiotropy and then screened the SNPs that met the requirements by linkage disequilibrium (LD) ($r^2 < 0.01$, kb > 5000). We also excluded SNPs with a minimum allele frequency (MAF) less than 0.01 because the impact of these SNPs was not stable. Palindromic variants with intermediate allele frequencies are also eliminated to prevent strand ambiguity errors. In addition, recent MR studies have found a causal relationship between short telomeres and hypothyroidism and the risk of IPF [20, 21]. By searching the PhenoScanner database [22], we removed the IVs strongly associated with hypothyroidism ($P < 5 \times 10^{-6}$), including rs3134605 related to HSV-1 IgG, and rs7452864, rs7745002 and rs74951723 related to EBNA1 IgG (S2 Table). No IVs related to short telomere strength were found. We also use F statistics [23] to avoid weak IVs bias. IVs with F-statistics greater than 10 are considered to have sufficient instrument strength. Finally, 22 SNPs associated with herpesvirus infections, 38 SNPs associated with circulating IgG levels, and 13 associated with mononucleosis and cold scores were used as IVs for the current MR analysis.

## Outcome GWAS data set

A large meta-analysis of IPF susceptibility was used for analysis [24]. This meta-analysis, based on the authors' previous independent case-control studies in the UK [25], Chicago [26], and Colorado [27], produced GWAS for the current maximum number of IPF cases (N = 11259, 2668 IPF patients and 8591 controls). The IPF cases in the three studies were diagnosed according to the American Thoracic Society/European Respiratory Society guidelines [28]. These studies have adopted strict quality control measures to select individuals of unrelated European origin. Each study's first ten principal components were adjusted to explain the population structure.

## Statistical analysis

After harmonizing the IVs from exposure and outcome through the same allele, we utilized a Wald ratio approach to assess the causality of one single IV to IPF [12]. We took the MR inverse variance-weighted (IVW) [29] as the primary approach to evaluate the impact of exposure on outcome. In the hypothesis of MR-IVW, intercept terms and the pleiotropy of IVs are not considered. Therefore, we implemented various sensitivity analysis to make up for the shortcomings of the IVW method. Even if nearly half of the instrumental SNPs are ineffective, the weighted median (WM) method could still provide consistent estimates [30]. The MR-Egger method can provide more conservative estimates of causal effects in the presence of horizontal pleiotropy and reduce the generation of exaggerated test results [31]. In addition, three heterogeneity tests were carried out. The Cochran's Q test was used to measure the heterogeneity between IVs. In the presence of heterogeneous results, we utilized the random-effects IVW approach to mitigate bias arising from weak IV associations [32, 33]. The MR-Egger

intercept detection considered the intercept term's existence and estimated the horizontal pleiotropy [34]. The MR-PRESSO approach detected the presence of outliers that may have horizontal pleiotropy through the global test [35]. In addition, we conducted leave-one-out test to remove individual IVs and calculated the MR results of the remaining IVs to identify the single SNP that is driving causality. The forest plot provided a visual representation of the impact of each SNP on the outcome. Finally, scatter plots were used to illustrate the fitting results of the various MR analyses.

As a measure of causal associations between herpesvirus-related exposures and the risk of IPF, we reported odds ratio (OR) and 95% confidence interval (CI) per one unit elevation in log-OR of diagnosed herpesvirus infection or one SD increase in relative circulating IgG levels. We use the network application mRND (https://shiny.cnsgenomics.com/ mRnd/) to calculate the statistical power of our MR analyses [36]. S3 Table showed the proportion of variant explained by IVs, F-statistics and power. The MR analysis of this study was carried out in the TwoSampleMR package [37] and the MR-PRESSO package [35] in R (Version 4.1.2). All statistical analyses showed bilateral distribution; P<0.05 was considered statistically significant. Bonferroni correction was applied to correct multiple comparisons. The threshold $p < 0.005$ (= 0.05/10 exposure) was considered a strong correlation. When the P value is between 0.005–0.05, we consider it to be a nominal association.

## Results

### Mendelian randomization estimates

S4 Table displayed the detailed information of SNPs associated with herpesvirus. The F statistics for 73 SNPs are greater than 20 (ranging from 21.92 to 51.47). Individual genetic estimates of each genetic variant were shown in S5 Table.

Fig 2 illustrated the effect of herpesvirus on the risk of IPF using the IVW method. Specifically, in the FinnGen cohort, genetically predicted EBV infection (OR = 1.105, 95%CI: 0.897–1.149, p = 0.815), CMV infection (OR = 1.073, 95%CI: 0.926–1.244, p = 0.302) and HSV

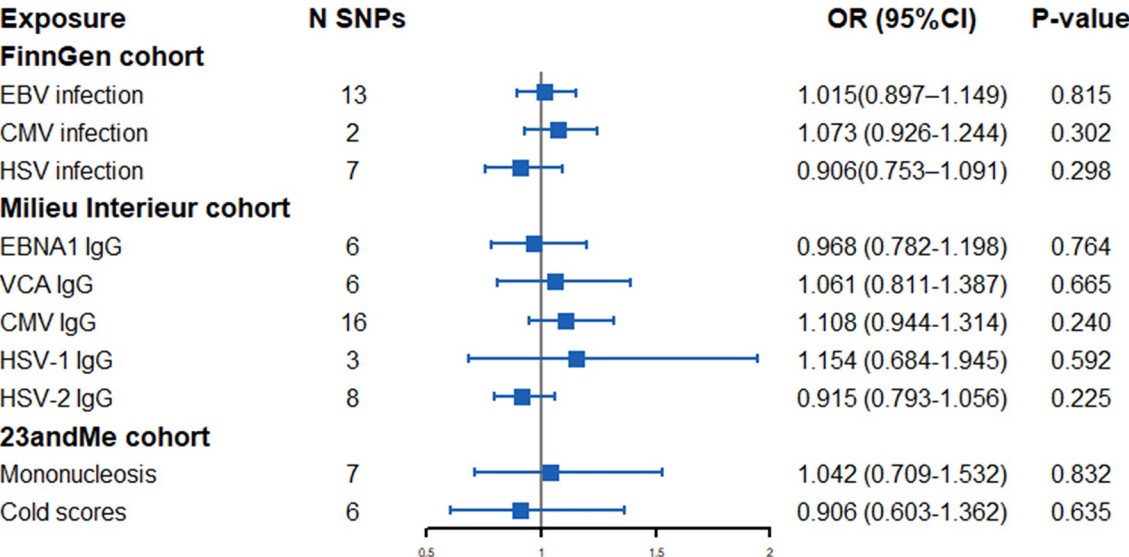

| Exposure | N SNPs | OR (95%CI) | P-value |
|---|---|---|---|
| **FinnGen cohort** | | | |
| EBV infection | 13 | 1.015(0.897–1.149) | 0.815 |
| CMV infection | 2 | 1.073 (0.926-1.244) | 0.302 |
| HSV infection | 7 | 0.906(0.753–1.091) | 0.298 |
| **Milieu Interieur cohort** | | | |
| EBNA1 IgG | 6 | 0.968 (0.782-1.198) | 0.764 |
| VCA IgG | 6 | 1.061 (0.811-1.387) | 0.665 |
| CMV IgG | 16 | 1.108 (0.944-1.314) | 0.240 |
| HSV-1 IgG | 3 | 1.154 (0.684-1.945) | 0.592 |
| HSV-2 IgG | 8 | 0.915 (0.793-1.056) | 0.225 |
| **23andMe cohort** | | | |
| Mononucleosis | 7 | 1.042 (0.709-1.532) | 0.832 |
| Cold scores | 6 | 0.906 (0.603-1.362) | 0.635 |

**Fig 2. Forest plot of Mendelian randomization estimates for association between each herpesvirus infection and idiopathic pulmonary fibrosis.** OR, odds ratio; CI, confidence interval; SNP, single nucleotide polymorphism; EBV, Epstein-Barr virus; CMV, cytomegalovirus; HSV, herpes simplex; EBNA1, EBV nuclear antigen-1; VCA, EBV viral capsid antigen; IgG, immunoglobulin G.

infection (OR = 0.906, 95%CI: 0.753–1.097, p = 0.298) were not associated with the risk of IPF. In the Milieu Intérieur cohort, we found no correlations between herpesvirus-related IgG EBNA1 (OR = 0.968, 95%CI: 0.782–1.198, p = 0.764), VCA (OR = 1.061, 95CI%: 0.811–1.387, p = 0.665), CMV (OR = 1.108, 95CI%: 0.944–1.314, p = 0.240), HSV-1 (OR = 1.154, 95%CI: 0.684–1.945, p = 0.592) and HSV-2 (OR = 0.915, 95%CI: 0.793–1.056, p = 0.225) and IPF risk. Additionally, in the 23andMe cohort, we also found no evidence of associations between mononucleosis (OR = 1.042, 95%CI: 0.709–1.532, p = 0.832) and cold scores (OR = 0.906, 95% CI: 0.603–1.362, p = 0.635) and IPF.

### Evaluation of Mendelian randomization assumptions

In sensitivity analysis, we only found a negative correlation between HSV infection (OR = 0.555, 95%CI: 0.364–0.846, p = 0.041) and IPF in the MR-Egger method. No evidence of associations between herpesviruses and IPF were found in other sensitivity analyses (S6 Table).

In heterogeneity test, Cochran's Q test found no heterogeneity between the instrumental SNP effects of the herpesvirus (S7 Table). We also found no evidence of horizontal pleiotropy in the MR-Egger intercept and MR-PRESSO global tests (S7 Table). Consistent with our previous test, leave-one-out analysis, forest plots and scatter plots proved that our findings are robust. (S1–S10 Figs).

## Discussion

This is the first MR analysis to investigate the causal association between three herpesvirus infections and IPF. Our results showed that three herpesvirus infections (EBV, CMV and HSV) and herpesvirus-related IgG levels (EBNA1, VCA, CMV, HSV-2 and HSV-2) were not significantly associated with IPF. Compared with previous observational studies, our results are less likely to be affected by confusion and reverse causality bias [38].

MR analysis is based on three hypotheses. First, there is a strong correlation between instrumental SNP and exposure. We selected the IVs for our study from three large GWAS. Although the proportion of variants explained by part IVs is not high in herpesvirus infections, the F-statistics of all instrumental SNPs is higher than 20, effectively avoiding weak IVs bias [23]. The other two hypotheses are related to pleiotropy. We searched the PhenoScanner database and excluded four SNPs associated with hypothyroidism. Next, the MR-Egger intercept and MR-PRESSO global tests were used to avoid horizontal pleiotropy. Therefore, our MR study's results are robust and unlikely to be affected by pleiotropy. In addition, most of the sensitivity analysis is consistent with the primary analysis (IVW method), which enhances the credibility of our findings.

The relationship between herpesvirus infection and IPF has been widely studied. Yonemaru et al. [6] found that the titers of VCA, CMV and HSV IgG in patients with IPF were significantly higher than in other lung diseases and normal controls. Tang et al. [39] detected the presence of eight herpesviruses in the lung tissues of 33 patients with PF (including 25 IPF). Their results showed that at least one herpes virus (EBV, CMV, HHV6 and 7) was detected in 97% of the patients, compared with 36% in the control group. Moreover, compared with other viruses, EBV is the dominant virus in IPF. Similar results have been obtained by Calabrese et al. [40]. However, some studies have yielded inconsistent results. Yin et al. [41] found no increase in herpesvirus gene transcription in 28 IPF lung transplant explants compared with 20 normal lung tissues by RNA next-generation sequencing. Le Hingrat et al. [42] found similar herpesvirus prevalence in 19 IPF groups and 9 control lung samples. It is worth noting that two recent meta-analyses have re-emphasized the link between herpesvirus and IPF. Sheng

et al. [7] implemented a meta-analysis including 20 case-control studies with a total of 1287 participants. They found that chronic infection of herpesviruses such as EBV, CMV, HHV7 and 8 significantly increased the risk of IPF. A recent meta-analysis [43] of 32 studies also found a possible link between viral infection and the pathogenesis of IPF. Among them, the highest infection rates were HSV (77.7% CI: 38.48%-99.32%) and EBV (72% CI: 44.65%-90.79%).

Nevertheless, most of the above studies come from retrospective case-control studies or meta-analyses. Some do not adjust for possibly crucial confounders like environmental exposure or smoking, hence distorting the real correlation between the exposure and disease. Meanwhile, different regions and races may also be the reasons for the different results of previous studies in a single region. Moreover, because of the low incidence of IPF, it is not easy to conduct large-scale prospective studies to investigate the actual causal relationship. Therefore, we utilized the MR design to evaluate the causal association between herpesvirus infections and IPF. Our results did not find a causal relationship between the three herpesvirus infections and related IgG levels and IPF risk. Therefore, herpesvirus infection may only be a phenomenon associated with lung injury and fibrosis.

The strengths of this study include that, in the absence of randomized controlled trials, MR is an economical and relatively reliable method for exploring the causal relationship between exposure and disease. Compared to observational studies, our results are less susceptible to the influence of confounders. Additionally, we selected three independent cohorts to assess the impact of herpesvirus infection on IPF. Consistent analysis results underscore the reliability of our findings.

Our study should also consider some limitations. First, our results need to be carefully interpreted. On one hand, one of the fundamental assumptions of MR is the strong correlation between IVs and exposure. In this study, despite using F-statistics to account for the influence of weak IVs, the association between the IVs and exposure may not be close because IVs were selected from different cohorts without clear overlap. On the other hand, conclusions drawn from MR studies can vary significantly. For instance, two MR studies examined the impact of gastro-esophageal reflux disease on IPF and arrived at completely different conclusions [44, 45]. This difference may be attributed to differences in the criteria for selecting IVs and the sample size of the exposure or outcome. Therefore, while our MR study provides preliminary evidence from a genetic perspective suggesting no causal relationship between herpesvirus infection and IPF, the possibility of such an association cannot be ruled out due to limitations in the strength of IVs and sample size. Further MR analysis will be needed when stronger IVs or GWAS with larger sample sizes become available.

Second, because IgG antibody levels only represent past herpesvirus infections, our results only indicate that previous herpesvirus infections are not associated with IPF. It cannot rule out the possibility of acute herpesvirus infections having an impact on IPF. Third, age is a known risk factor for IPF. The GWAS summary statistics rather than individual-level data limit our assessment of the causal relationship between herpesvirus infection and IPF risk of different ages and genders. Fourth, the significance of many SNPs in other diseases remains to be explored; therefore, although our pleiotropy analysis excludes some of the SNPs related to hypothyroidism, it is undeniable that unknown, polymorphic SNPs may still affect our results. Finally, the GWAS data in our study came from the European populations. Therefore, our results may not generalize to other populations. It is necessary to conduct a multi-population MR study in the future to verify our findings.

In conclusion, this study provides preliminary evidence that EBV, CMV, and HSV herpesviruses, and their related IgG levels, are not causally linked to IPF. However, our conclusions should be interpreted carefully. Further MR analysis will be necessary when stronger IVs and GWAS with larger sample sizes become available.

## Supporting information

**S1 Fig. The leave-one-out plot, forest plot, and scatter plot for the association of EBV infection and idiopathic pulmonary fibrosis.**
(DOCX)

**S2 Fig. The forest plot, and scatter plot for the association of CMV infection and idiopathic pulmonary fibrosis.**
(DOCX)

**S3 Fig. The leave-one-out plot, forest plot, and scatter plot for the association of HSV infection and idiopathic pulmonary fibrosis.**
(DOCX)

**S4 Fig. The leave-one-out plot, forest plot, and scatter plot for the association of EBNA1 IgG level and idiopathic pulmonary fibrosis in discovery analysis.**
(DOCX)

**S5 Fig. The leave-one-out plot, forest plot, and scatter plot for the association of VCA IgG level and idiopathic pulmonary fibrosis in discovery analysis.**
(DOCX)

**S6 Fig. The leave-one-out plot, forest plot, and scatter plot for the association of CMV IgG level and idiopathic pulmonary fibrosis in discovery analysis.**
(DOCX)

**S7 Fig. The leave-one-out plot, forest plot, and scatter plot for the association of HSV-1 IgG level and idiopathic pulmonary fibrosis in discovery analysis.**
(DOCX)

**S8 Fig. The leave-one-out plot, forest plot, and scatter plot for the association of HSV-2 IgG level and idiopathic pulmonary fibrosis in discovery analysis.**
(DOCX)

**S9 Fig. The leave-one-out plot, forest plot, and scatter plot for the association of mononucleosis and idiopathic pulmonary fibrosis.**
(DOCX)

**S10 Fig. The leave-one-out plot, forest plot, and scatter plot for the association of cold cores and idiopathic pulmonary fibrosis.**
(DOCX)

**S1 Table. Brief description of datasets utilized in the Mendelian randomization study.**
(DOCX)

**S2 Table. Association of potential pleiotropic SNPs searched in the Phenoscanner database.**
(DOCX)

**S3 Table. Statistical power for the Mendelian randomization analyses of herpesvirus infection or herpesvirus infection-related IgG level and risk of IPF.**
(DOCX)

**S4 Table. Instrumental variables for genetically predicted herpesviruses infection or herpesvirus infection-related IgG level in the Mendelian randomization analysis.**
(DOCX)

**S5 Table. Summary information for SNPs that were used as genetic instruments for Mendelian randomization analyses of genetically predicted herpesviruses infection or herpesvirus-related IgG levels and risk of IPF.**
(DOCX)

**S6 Table. Association of SNPs for herpesvirus infection or herpesvirus infection-related IgG level with IPF using MR with different methods.**
(DOCX)

**S7 Table. Cochran's Q test, MR-Egger intercept and MR-PRESSO Mendelian randomization analyses of herpesvirus infection or herpesvirus infection-related IgG level and risk of IPF.**
(DOCX)

## Acknowledgments

We want to acknowledge the participants and investigators of FinnGen study and the Milieu Intérieur Consortium. We authors are grateful to them for making summary-level association statistics of herpesvirus infection and herpesvirus-related IgG levels possible and accessible. We also would like to thank Collaborative Group of genetic studies of IPF for providing us with the IPF GWAS summary data.

## Author Contributions

**Conceptualization:** Haihao Yan, Chenghua Zhu, Ganzhu Feng.

**Methodology:** Xiao Jin.

**Software:** Xiao Jin.

**Validation:** Haihao Yan, Chenghua Zhu.

**Visualization:** Haihao Yan.

**Writing – original draft:** Haihao Yan, Chenghua Zhu.

**Writing – review & editing:** Ganzhu Feng.

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
