## [Decision Letter · Decision Letter 0]

14 Sep 2023

PONE-D-23-25486Mendelian randomization reveals no correlations between herpesvirus infection and idiopathic pulmonary fibrosisPLOS ONE

Dear Dr. Feng,

Thank you for submitting your manuscript to PLOS ONE. After careful consideration, we feel that it has merit but does not fully meet PLOS ONE’s publication criteria as it currently stands. Therefore, we invite you to submit a revised version of the manuscript that addresses the points raised during the review process.

We look forward to receiving your revised manuscript.

Kind regards,

Simone Agostini, Ph.D.

Academic Editor

PLOS ONE

2. Please remove your figures from within your manuscript file, leaving only the individual TIFF/EPS image files, uploaded separately. These will be automatically included in the reviewers’ PDF.

3. We are unable to open your Supporting Information file [Supplementary materials.docx]. Please kindly revise as necessary and re-upload.

5. We noticed you have some occurrence of overlapping text with the following previous publication(s), which needs to be addressed:

-https://www.mdpi.com/2072-6643/14/21/4569/htm

In your revision ensure you cite all your sources (including your own works), and quote or rephrase any duplicated text outside the methods section. Further consideration is dependent on these concerns being addressed.

Reviewers' comments:

Reviewer's Responses to Questions

**Comments to the Author**

1. Is the manuscript technically sound, and do the data support the conclusions?

Reviewer #1: Partly

Reviewer #2: Partly

2. Has the statistical analysis been performed appropriately and rigorously? 

Reviewer #1: I Don't Know

Reviewer #2: No

3. Have the authors made all data underlying the findings in their manuscript fully available?

Reviewer #1: Yes

Reviewer #2: Yes

4. Is the manuscript presented in an intelligible fashion and written in standard English?

Reviewer #1: Yes

Reviewer #2: Yes

5. Review Comments to the Author

Reviewer #1: In their manuscript, the authors employ Mendelian randomization (MR) analysis on previously published GWAS data to suggest that herpesvirus infection does not causally influence the onset of idiopathic pulmonary fibrosis (IPF). While I may not possess the comprehensive expertise to delve deep into the analytical methods, I have two primary points of feedback:

The strength of association between the chosen SNPs, which serve as instrumental variables (IVs), and the exposures (i.e., herpes viral infections) remains unclear. A core premise of MR studies is the tight correlation between IVs and exposure. From the presented data, the selected IVs seem to exhibit a weak relationship with herpes infection/IgG levels. Should there be a strong association, one would expect to observe the same SNPs consistently identified across diverse study cohorts. However, no such overlap is evident among IVs from different cohorts, even though each has its distinct criteria for determining herpesvirus infection. Given the high prevalence of herpesvirus infections in humans, it seems improbable to pinpoint SNPs significantly associated with these infections.

As the study concludes with a negative correlation, it might be beneficial to apply the same MR method to investigate a known positive risk factor for IPF, such as smoking. Demonstrating such a correlation would greatly bolster the credibility of the findings and enhance reader confidence.

Reviewer #2: Dear authors,

There are some issues regarding the methodology of this article. First of all instrumental variable is not explained adequately and some statistical terms are not used properly such as hypothesis instead of assumption or using of confusion instead of confounding.

Moreover, the selection of the gene used for instrumental variable is not explained. I recommend to consult a methodologist to recheck this area.

6. PLOS authors have the option to publish the peer review history of their article (what does this mean?). If published, this will include your full peer review and any attached files.

Reviewer #1: No

Reviewer #2: No

---

## [Author Response · Author response to Decision Letter 0]

28 Sep 2023

Dear editor,

Thank you for giving us the opportunity to submit a revised draft of the manuscript “Mendelian randomization reveals no correlations between herpesvirus infection and idiopathic pulmonary fibrosis” for publication in the Journal of PLOS ONE. We appreciate the time and effort that you and the reviewers dedicated to providing feedback on our manuscript and are grateful for the insightful comments on and valuable improvements to our paper. Based on your suggestions, we have modified the format of the article and repeated content. I hope that the changes I’ve made resolve all your concerns about the article. I’m more than happy to make any further changes that will improve the paper and facilitate successful publication.

Reviewer 1

Dear Reviewer,

Thanks very much for taking your time to review this manuscript. I really appreciate your comments and suggestions! Please see below, for a point-by-point response to the reviewer’s comments and concerns. I hope that the changes I’ve made resolve all your concerns about the article. I’m more than happy to make any further changes that will improve the paper and facilitate successful publication.

Point 1: The strength of association between the chosen SNPs, which serve as instrumental variables (IVs), and the exposures (i.e., herpes viral infections) remains unclear. A core premise of MR studies is the tight correlation between IVs and exposure. From the presented data, the selected IVs seem to exhibit a weak relationship with herpes infection/IgG levels. Should there be a strong association, one would expect to observe the same SNPs consistently identified across diverse study cohorts. However, no such overlap is evident among IVs from different cohorts, even though each has its distinct criteria for determining herpesvirus infection. Given the high prevalence of herpesvirus infections in humans, it seems improbable to pinpoint SNPs significantly associated with these infections.

Response 1: Thank you for pointing this out. Indeed, there is a relatively weak association between some selected IVs and herpesvirus infection/IgG levels. Considering the high prevalence of herpesvirus infection in humans, it is challenging to pinpoint SNPs significantly correlated with these infections. However, to address this limitation, we opted for three independent cohorts to validate the impact of herpesvirus infection on IPF onset, thereby enhancing the credibility of our results. It is worth noting that the herpesvirus infection-related data used in this study have been employed in several MR studies. This suggests that utilizing these data for herpesvirus-related MR analysis is reasonably justified to some extent. Specific papers for reference are as follows:

1. Zhang W, Wu P, Yin R, et al. Mendelian Randomization Analysis Suggests No Associations of Herpes Simplex Virus Infections With Multiple Sclerosis. Front Neurosci. 2022;16:817067. Published 2022 Mar 1. doi:10.3389/fnins.2022.817067.

2. Huang SY, Yang YX, Kuo K, et al. Herpesvirus infections and Alzheimer's disease: a Mendelian randomization study. Alzheimers Res Ther. 2021;13(1):158. Published 2021 Sep 24. doi:10.1186/s13195-021-00905-5.

3. Zhang Y, Qu J, Luo L, Xu Z, Zou X. Multigenomics Reveals the Causal Effect of Herpes Simplex Virus in Alzheimer's Disease: A Two-Sample Mendelian Randomization Study. Front Genet. 2022;12:773725. Published 2022 Jan 5. doi:10.3389/fgene.2021.773725.

4. Tan JS, Ren JM, Fan L, et al. Genetic Predisposition of Anti-Cytomegalovirus Immunoglobulin G Levels and the Risk of 9 Cardiovascular Diseases. Front Cell Infect Microbiol. 2022;12:884298. Published 2022 Jun 27. doi:10.3389/fcimb.2022.884298.

5. Zhu G, Zhou S, Xu Y, et al. Chickenpox and multiple sclerosis: A Mendelian randomization study. J Med Virol. 2023;95(1):e28315. doi:10.1002/jmv.28315.

Point 2: As the study concludes with a negative correlation, it might be beneficial to apply the same MR method to investigate a known positive risk factor for IPF, such as smoking. Demonstrating such a correlation would greatly bolster the credibility of the findings and enhance reader confidence.

Response 2: We have identified an existing MR study that investigated the causal relationship between smoking and IPF. This study revealed that the genetic predisposition to initiate smoking (based on 378 variants) and lifelong smoking (based on 126 variants) is associated with a higher risk of IPF. The results of this study also demonstrate the reliability of using MR to explore risk factors for IPF. Specific paper for reference is as follows:

1. Zhu J, Zhou D, Yu M, Li Y. Appraising the causal role of smoking in idiopathic pulmonary fibrosis: a Mendelian randomization study [published online ahead of print, 2023 May 22]. Thorax. 2023;thorax-2023-220012. doi:10.1136/thorax-2023-220012.

Reviewer 2

Dear Reviewer,

Thanks very much for taking your time to review this manuscript. I really appreciate your comments and suggestions! Please see below, for a point-by-point response to the reviewer’s comments and concerns. I hope that the changes I’ve made resolve all your concerns about the article. I’m more than happy to make any further changes that will improve the paper and facilitate successful publication.

Point 1: There are some issues regarding the methodology of this article. First of all instrumental variable is not explained adequately and some statistical terms are not used properly such as hypothesis instead of assumption or using of confusion instead of confounding.

Response 1: Based on your advice, we have corrected the misuse of statistical terms and provided a more detailed explanation of the concept of instrumental variables. Please see page 7, lines 73-82 for details.

Point 2: Moreover, the selection of the gene used for instrumental variable is not explained. I recommend to consult a methodologist to recheck this area.

Response 2: Indeed, confirming the genetic selection for instrumental variables can enhance the credibility of the chosen instrumental variables. However, based on the current GWAS data related to herpesvirus infection, we were unable to determine the necessary genes for instrumental variable selection. It is worth noting that the selection process for IVs in this paper was based on a substantial body of previously published Mendelian Randomization studies associating herpesvirus infection with diseases. These prior publications, to some extent, can substantiate the rationale and reliability of our instrumental variable selection. Specific papers for reference are as follows:

1. Zhang W, Wu P, Yin R, et al. Mendelian Randomization Analysis Suggests No Associations of Herpes Simplex Virus Infections With Multiple Sclerosis. Front Neurosci. 2022;16:817067. Published 2022 Mar 1. doi:10.3389/fnins.2022.817067.

2. Huang SY, Yang YX, Kuo K, et al. Herpesvirus infections and Alzheimer's disease: a Mendelian randomization study. Alzheimers Res Ther. 2021;13(1):158. Published 2021 Sep 24. doi:10.1186/s13195-021-00905-5.

3. Zhang Y, Qu J, Luo L, Xu Z, Zou X. Multigenomics Reveals the Causal Effect of Herpes Simplex Virus in Alzheimer's Disease: A Two-Sample Mendelian Randomization Study. Front Genet. 2022;12:773725. Published 2022 Jan 5. doi:10.3389/fgene.2021.773725.

4. Tan JS, Ren JM, Fan L, et al. Genetic Predisposition of Anti-Cytomegalovirus Immunoglobulin G Levels and the Risk of 9 Cardiovascular Diseases. Front Cell Infect Microbiol. 2022;12:884298. Published 2022 Jun 27. doi:10.3389/fcimb.2022.884298.

5. Zhu G, Zhou S, Xu Y, et al. Chickenpox and multiple sclerosis: A Mendelian randomization study. J Med Virol. 2023;95(1):e28315. doi:10.1002/jmv.28315.

---

## [Decision Letter · Decision Letter 1]

31 Oct 2023

PONE-D-23-25486R1Mendelian randomization reveals no correlations between herpesvirus infection and idiopathic pulmonary fibrosisPLOS ONE

Dear Dr. Feng,

Thank you for submitting your manuscript to PLOS ONE. After careful consideration, we feel that it has merit but does not fully meet PLOS ONE’s publication criteria as it currently stands. Therefore, we invite you to submit a revised version of the manuscript that addresses the points raised during the review process.

We look forward to receiving your revised manuscript.

Kind regards,

Simone Agostini, Ph.D.

Academic Editor

PLOS ONE

Journal Requirements:

2. We noticed you have some occurrence of overlapping text with the following previous publication(s), which needs to be addressed:

-https://www.mdpi.com/2072-6643/14/21/4569/htm

In your revision ensure you cite all your sources (including your own works), and quote or rephrase any duplicated text outside the methods section. Further consideration is dependent on these concerns being addressed.

Reviewers' comments:

Reviewer's Responses to Questions

**Comments to the Author**

1. If the authors have adequately addressed your comments raised in a previous round of review and you feel that this manuscript is now acceptable for publication, you may indicate that here to bypass the “Comments to the Author” section, enter your conflict of interest statement in the “Confidential to Editor” section, and submit your "Accept" recommendation.

Reviewer #1: (No Response)

2. Is the manuscript technically sound, and do the data support the conclusions?

Reviewer #1: Yes

3. Has the statistical analysis been performed appropriately and rigorously? 

Reviewer #1: I Don't Know

4. Have the authors made all data underlying the findings in their manuscript fully available?

Reviewer #1: Yes

5. Is the manuscript presented in an intelligible fashion and written in standard English?

Reviewer #1: Yes

6. Review Comments to the Author

Reviewer #1: The current revision does not adequately address my concerns. Given the complexity and limitations inherent to the study, I believe there's a need for revisions in the discussion and conclusion sections:

1. The observed weak correlation between the instrumental variables (IVs) and herpes infection merits a comprehensive discussion in the limitations section. It is vital to note that a strong correlation between IVs and exposure is a foundational assumption for Mendelian randomization (MR) studies.

2. It's crucial to acknowledge that conclusions derived from MR studies can vary significantly or even contradict each other. For instance, two MR studies examining the causative effects of smoking on Idiopathic Pulmonary Fibrosis (IPF) – Zhu et al., 2023 in Thorax, and Duckworth et al., 2020 in medRxiv – reached contrasting conclusions. Such differences could arise from the choice of IVs or the sample size of IPF patients. Highlighting these potential pitfalls is essential for reader awareness.

To summarize, while MR studies offer a novel perspective for exploring the causal link between herpesvirus infections and IPF through a genetic perspective, I urge a more cautious presentation of the conclusions. The absence of evidence for a causal relationship between an exposure and a disease, particularly when using a limited measure, doesn't negate the potential for such a relationship. This is especially pertinent when there is no positive control to validate the methodology employed.

7. PLOS authors have the option to publish the peer review history of their article (what does this mean?). If published, this will include your full peer review and any attached files.

Reviewer #1: No

---

## [Author Response · Author response to Decision Letter 1]

7 Nov 2023

Dear editor,

Thank you for giving us the opportunity to submit a revised draft of the manuscript “Mendelian randomization reveals no correlations between herpesvirus infection and idiopathic pulmonary fibrosis” for publication in the Journal of PLOS ONE. We appreciate the time and effort that you and the reviewers dedicated to providing feedback on our manuscript and are grateful for the insightful comments on and valuable improvements to our paper. According to your suggestion, we revised the repetition in the article and revised the misquoted references. Specifically, we modify the following references

1. Widener, R. W. and Whitley, R. J. Herpes simplex virus. Handb Clin Neurol. 2014, 123,251-63.10.1016/b978-0-444-53488-0.00011-0.; 

2. Epstein, M. A. Aspects of the EB virus. Adv Cancer Res. 1970, 13,383-411.10.1016/s0065-230x(08)60169-4; Zhang, L. J.; Hanff, P.; Rutherford, C.; 

3. Churchill, W. H. and Crumpacker, C. S. Detection of human cytomegalovirus DNA, RNA, and antibody in normal donor blood. J Infect Dis. 1995, 171,1002-6.10.1093/infdis/171.4.1002.

to: 

DUCKWORTH, H.J. LONGHURST, J.K. PAXTON, et al. "The Role of Herpes Viruses in Pulmonary Fibrosis," Front Med (Lausanne),8: 704222,2021. 

I hope that the changes I’ve made resolve all your concerns about the article. I’m more than happy to make any further changes that will improve the paper and facilitate successful publication.

Reviewer 1

Dear Reviewer,

Thanks very much for taking your time to review this manuscript. I really appreciate your comments and suggestions! Please see below, for a point-by-point response to the reviewer’s comments and concerns. I hope that the changes I’ve made resolve all your concerns about the article. I’m more than happy to make any further changes that will improve the paper and facilitate successful publication.

Point 1: The observed weak correlation between the instrumental variables (IVs) and herpes infection merits a comprehensive discussion in the limitations section. It is vital to note that a strong correlation between IVs and exposure is a foundational assumption for Mendelian randomization (MR) studies.

Response 1: We thank the reviewer for this constructive suggestion. According to your suggestion, we have had a comprehensive discussion in the limitation section of the Discussion:

[Discussion: “First, our results need to be carefully interpreted. On one hand, one of the fundamental assumptions of MR is the strong correlation between IVs and exposure. In this study, despite using F-statistics to account for the influence of weak IVs, the association between the IVs and exposure may not be close because IVs were selected from different cohorts without clear overlap……Therefore, while our MR study provides preliminary evidence from a genetic perspective suggesting no causal relationship between herpesvirus infection and IPF, the possibility of such an association cannot be ruled out due to limitations in the strength of IVs and sample size. Further MR analysis will be needed when stronger IVs or GWAS with larger sample sizes become available.…”] (page 10-11, lines 281-294)

Point 2: It's crucial to acknowledge that conclusions derived from MR studies can vary significantly or even contradict each other. For instance, two MR studies examining the causative effects of smoking on Idiopathic Pulmonary Fibrosis (IPF) – Zhu et al., 2023 in Thorax, and Duckworth et al., 2020 in medRxiv – reached contrasting conclusions. Such differences could arise from the choice of IVs or the sample size of IPF patients. Highlighting these potential pitfalls is essential for reader awareness.

Response 2: Thank you for pointing out the problem and your valuable comments. We have highlighted this difference in the Discussion. Since the MR study published by Duckworth et al. was not published in a journal included in SCI, we chose to use two MR studies on the association between gastroesophageal reflux disease and IPF with different conclusions to demonstrate this difference.

[Discussion: “On the other hand, conclusions drawn from MR studies can vary significantly. For instance, two MR studies examined the impact of gastro-esophageal reflux disease on IPF and arrived at completely different conclusions [44, 45]. This difference may be attributed to differences in the criteria for selecting IVs and the sample size of the exposure or outcome. Therefore, while our MR study provides preliminary evidence from a genetic perspective suggesting no causal relationship between herpesvirus infection and IPF, the possibility of such an association cannot be ruled out due to limitations in the strength of IVs and sample size. Further MR analysis will be needed when stronger IVs or GWAS with larger sample sizes become available……”] (page 10-11, lines 286-294)

Point 3: To summarize, while MR studies offer a novel perspective for exploring the causal link between herpesvirus infections and IPF through a genetic perspective, I urge a more cautious presentation of the conclusions. The absence of evidence for a causal relationship between an exposure and a disease, particularly when using a limited measure, doesn't negate the potential for such a relationship. This is especially pertinent when there is no positive control to validate the methodology employed.

Response 3: Thank you for your suggestion. At your request, we have revised the Discussion and Conclusions.

[Discussion: “Therefore, while our MR study provides preliminary evidence from a genetic perspective suggesting no causal relationship between herpesvirus infection and IPF, the possibility of such an association cannot be ruled out due to limitations in the strength of IVs and sample size. Further MR analysis will be needed when stronger IVs or GWAS with larger sample sizes become available……”] (page 11, lines 290-294)

[Conclusions: “In conclusion, this study provides preliminary evidence that EBV, CMV, and HSV herpesviruses, and their related IgG levels, are not causally linked to IPF. However, our conclusions should be interpreted carefully. Further MR analysis will be necessary when stronger IVs and GWAS with larger sample sizes become available……”] (page 11, lines 307-310)

---

## [Decision Letter · Decision Letter 2]

15 Nov 2023

Mendelian randomization reveals no correlations between herpesvirus infection and idiopathic pulmonary fibrosis

PONE-D-23-25486R2

Dear Dr. Feng,

We’re pleased to inform you that your manuscript has been judged scientifically suitable for publication and will be formally accepted for publication once it meets all outstanding technical requirements.

Kind regards,

Simone Agostini, Ph.D.

Academic Editor

PLOS ONE

Additional Editor Comments (optional):

Reviewers' comments:

Reviewer's Responses to Questions

**Comments to the Author**

1. If the authors have adequately addressed your comments raised in a previous round of review and you feel that this manuscript is now acceptable for publication, you may indicate that here to bypass the “Comments to the Author” section, enter your conflict of interest statement in the “Confidential to Editor” section, and submit your "Accept" recommendation.

Reviewer #1: All comments have been addressed

2. Is the manuscript technically sound, and do the data support the conclusions?

Reviewer #1: (No Response)

3. Has the statistical analysis been performed appropriately and rigorously? 

Reviewer #1: (No Response)

4. Have the authors made all data underlying the findings in their manuscript fully available?

Reviewer #1: (No Response)

5. Is the manuscript presented in an intelligible fashion and written in standard English?

Reviewer #1: (No Response)

6. Review Comments to the Author

Reviewer #1: (No Response)

7. PLOS authors have the option to publish the peer review history of their article (what does this mean?). If published, this will include your full peer review and any attached files.

Reviewer #1: No

---

## [Editor Report · Acceptance letter]

16 Nov 2023

PONE-D-23-25486R2 

Mendelian randomization reveals no correlations between herpesvirus infection and idiopathic pulmonary fibrosis 

Dear Dr. Feng:

I'm pleased to inform you that your manuscript has been deemed suitable for publication in PLOS ONE. Congratulations! Your manuscript is now with our production department. 

Kind regards, 

on behalf of

Dr. Simone Agostini 

Academic Editor

PLOS ONE